# A Novel Method for the Fabrication of Antibacterial Stainless Steel with Uniform Silver Dispersions by Silver Nanoparticle/Polyethyleneimine Composites

**DOI:** 10.3390/ma16103719

**Published:** 2023-05-14

**Authors:** Yu-Kun Chih, Jhu-Lin You, Wei-Hsuan Lin, Yen-Hao Chang, Chun-Chieh Tseng, Ming-Der Ger

**Affiliations:** 1Graduate School of Defense Science, Chung Cheng Institute of Technology, National Defense University, Taoyuan 335, Taiwan; chihyk@gmail.com; 2Department of Chemical and Materials Engineering, Chung Cheng Institute of Technology, National Defense University, Taoyuan 335, Taiwan; yolin1014001@gmail.com (J.-L.Y.); j0939733557@gmail.com (W.-H.L.); 3System Engineering and Technology Program, National Chiao Tung University, Hsinchu 300, Taiwan; 4Combination Medical Device Technology Division, Medical Devices and Opto-Electronics Equipment Department, Metal Industries Research & Development Centre, Kaohsiung 821, Taiwan

**Keywords:** antibacterial stainless steel, silver nanoparticles (NPs), dispersions, polyethyleneimine, sintering

## Abstract

Only a few studies have so far focused on the addition of silver to SS316L alloys by conventional sintering methods. Unfortunately, the metallurgical process of silver-containing antimicrobial SS is greatly limited due to the extremely low solubility of silver in iron and its tendency to precipitate at the grain boundaries, resulting in an inhomogeneous distribution of the antimicrobial phase and loss of antimicrobial properties. In this work, we present a novel approach to fabricate antibacterial stainless steel 316L by functional polyethyleneimine-glutaraldehyde copolymer (PEI-co-GA/Ag catalyst) composites. PEI is a highly branched cationic polymer, which makes it exhibit very good adhesion on the surface of the substrate. Unlike the effect of the conventional silver mirror reaction, the introduction of functional polymers can effectively improve the adhesion and distribution of Ag particles on the surface of 316LSS. It can be seen from the SEM images that a large number of silver particles are retained and well dispersed in 316LSS after sintering. PEI-co-GA/Ag 316LSS exhibits excellent antimicrobial properties and does not release free silver ions to affect the surrounding environment. Furthermore, the probable mechanism for the influence of the functional composites on the enhancement of adhesion is also proposed. The formation of a large number of hydrogen bonds and van der Waals forces, as well as the negative zeta potential of the 316LSS surface, can effectively enable the formation of a tight attraction between the Cu layer and the surface of 316LSS. These results meet our expectations of designing passive antimicrobial properties on the contact surface of medical devices.

## 1. Introduction

Stainless steel (SS) has attracted much attention in recent years due to its numerous excellent properties, such as corrosion resistance, high strength, good biocompatibility, and superior workability, especially as a common material in the field of public health. Stainless steel 316L (316LSS) is one of the most widely used materials in biomedical applications because of its high-strain properties, non-magnetic and low cost [1,2,3]. In the medical field, it is estimated that approximately 60% of surgical implants and about 85% of surgical instruments used in the United States are made of SS [4,5]. However, SS is not inherently antibacterial; the bacteria will attach to the metal surface and proliferate to form a bacterial biofilm, which may lead to contamination of the SS surface and increase the risk of infection, which has significant harmful effects on humans [6,7,8]. Thus, developing novel antimicrobial SSs and enhancing their antimicrobial properties will play a key role in using biomedical devices and preventing biomedical device-related infections [9,10,11,12].

Many studies have shown that Ag, Au, Cu, Zn, and Mg can destroy the cell membrane and genetic material of bacteria and further inhibit the reproduction of bacteria [13,14,15,16,17,18]. Silver (Ag) is considered the most promising due to its excellent antimicrobial property, even in small amounts, which is about 100 times stronger than Cu [19,20]. However, owing to the extremely low solubility of Ag in Fe and its easy precipitation at grain boundaries [21], the metallurgical process of Ag-containing antibacterial SS is greatly limited, resulting in an uneven distribution of the antibacterial phase and loss of antibacterial properties. Currently, most studies related to obtaining antimicrobial SSs are focused on surface engineering, such as surface coating [22], plasma spraying [23], ion implantation [24], liquid flame spraying [25], spraying deposition [26], and electrodeposition [27], etc. Nonetheless, these methods must overcome several problems, the most difficult of which is poor adhesion between the coating and the substrate [28]. Namely, the retention of Ag particles deposited on the surface during application is still being determined. Ag-SS alloys are another effective way to improve antibacterial performance. They can compensate for the surface treatment technology’s shortcomings that only allow doping in the outermost layer (a few nanometers) and improve the chances of Ag particles remaining on the surface. However, since the dispersion of Ag particles is not well controlled [29,30,31,32], up to now, only a few studies have focused on the addition of Ag to the SS316L alloy by the conventional sintering method. Thus, developing a simple and versatile strategy for enhancing adhesion between Ag particles and SS and the uniform distribution of Ag particles on the SS matrix is challenging.

In our previous studies [33,34], we successfully developed a variety of nanocomposite catalysts, which acted as active nuclei on the surface of different materials during electroless plating, effectively enhancing the adhesion of metals on substrates. This method overcomes the extremely low solubility of silver in iron and its precipitation at grain boundaries and further achieves a uniform dispersion of silver particles on the surface of 316LSS. However, we have yet to attempt electroless silver plating on stainless steel surfaces using composite catalysts. This work reports a facile and reliable method for preparing Ag-containing antibacterial SS. The 316LSS powder was used as the raw material. 316LSS was simply dipped into a functional polyethyleneimine-glutaraldehyde copolymer (PEI-co-GA/Ag catalyst), and then Ag-containing antibacterial 316LSS was prepared by electroless plating. The nanoscale Ag-rich particles were uniformly formed in the SS matrix without agglomeration during the subsequent sintering process. This is attributed to the fact that various functional groups in the structure of the PEI-co-GA/Ag catalyst can assist dispersion in chelating the Ag nanoparticles (Ag NPs), effectively preventing them from aggregating [35,36,37]. In addition, Ag nanoparticles can be uniformly retained in the 316LSS alloy with proper sintering parameters.

Consequently, we have innovatively developed a technology to deposit Ag particles on the surface of 316LSS through electroless plating, which also applies to the subsequent sintering process. The composition of the Ag-containing antimicrobial 316LSS and its internal Ag distribution were investigated by scanning electron microscopy (SEM) and EDS element analysis. The probable mechanism for the influence of the functional composites on the enhancement of adhesion is further proposed. The bacterial inhibition tests of Ag-containing antibacterial 316LSS are also discussed.

## 2. Material and Methods

### 2.1. Preparation of Functionalized PEI-co-GA/Ag Catalyst

All chemicals in this work were of analytic grade purity and acquired from Merck. To begin with, PEI-GA copolymers (PEI-co-GA) were prepared by copolymerizing 5 mL of 1 wt% of a polyethyleneimine (PEI, Mw 10000, Alfa Aesar, Haverhill, MA, USA) aqueous solution and 5 mL of glutaraldehyde (50% GA, C_5_H_8_O_2_, Alfa Aesar, Haverhill, MA, USA) at room temperature for 30 min. This step aims to achieve some degree of cross-linking between PEI and GA. Then, the PEI-co-GA aqueous solution and the silver nitrate aqueous solution (AgNO_3_, Echo Chemical, 200 mg dissolved in 190 mL of deionized water) were mixed at 90 °C for 10 min to obtain a PEI-co-GA/Ag catalyst with a concentration of 600 PPM. Finally, the synthesized PEI-co-GA/Ag catalyst was dialyzed against deionized water for 12 h to remove the unreacted reactants.

### 2.2. Electroless Silver Plating, Mirror Silver Reaction and Sintering

The experimental process is shown in Figure 1. First, 100 g of 316LSS powder was immersed into the PEI-co-GA/Ag catalyst and stirred at 300 rpm at 60 °C to fully contact the metal powder with the catalyst. Secondly, the 316LSS powder containing the catalyst was filtered and dried in a vacuum at 90 °C, then immersed in an electroless plating bath for electroless silver plating. The electroless bath for Ag contains silver nitrate (AgNO_3_ 2 g), ammonia aqueous (NH_4_OH 15 mL), and sodium hydroxide (1M NaOH 0.5 mL) in 200 mL of deionized water. Next, formaldehyde was added to the electroless bath as a reducing agent. It is worth noting that, in general, the silver mirror reaction is required for more formaldehyde (0.8 mL in the study) to carry out the reduction reaction. However, in the case of functional polymers, the reduction of silver particles can be achieved by lower amounts of formaldehyde. In the study, electroless silver plating was performed by the dropwise addition of formaldehyde (0.4 mL) at room temperature with a reaction time of 4 h. Finally, the Ag-containing 316LSS powder was obtained by filtering and drying under a vacuum at 80 °C for 8 h. In this study, the parameters of sintering, as a process to consolidate powders, were also carried out under operating conditions (at a temperature of around 1100 °C for 200 min in 10^−2^ Torr) to obtain an optimal product. As shown in Figure 2, the Ag-containing 316LSS powder was molded at 300 N/m^2^ to a dimension of 11 mm in diameter and 5 mm in thickness. Finally, the optimized Ag-containing antibacterial 316LSS was obtained by a sintering process.

### 2.3. Characterizations

The scanning electron micrographs (SEMs) of the Ag-containing 316LSSs were recorded on a Hitachi S-3000 N operating at 10 kV in secondary electron imaging mode. The setup was also equipped with an energy-dispersive X-ray (EDX) detection system and a mapping function. Besides that, the pilot studies of an antibacterial test by two different methods were also performed.

### 2.4. Evaluating Methods of Antibacterial Activity

There are various methods used to evaluate the antibacterial activity of stainless steel [38,39]; in the pilot study, we used the Kirby–Bauer Disk Diffusion Test [40] and Colony Forming Units [41], which are two commonly used methods, to evaluate the antibacterial activity of different stainless steels.

## 3. Results and Discussion

### 3.1. Characterization of Chemical Silver Plating on Stainless Steel Powders without the Use of Functional Polymers

As shown by the SEM image of the 316LSS powders in Figure 3a, these powders have a random particle size of about 2–7 μm. The surface morphology of the powder in Figure 3b shows that the surface of the powder was very smooth and flat, which may not be conducive to the deposition of Ag particles. In general, the mirror silver reaction is the most common metallization method. To clarify the intrinsic interaction (the adhesion between the two metals) between the 316LSS powder and the silver particles, the adhesion of Ag on the 316LSS surface was first observed by the silver mirror reaction without the use of functional polymers. Figure 4a clearly shows that only a small amount of Ag particles was attached to the surface of 316LSS, with the non-uniform distribution (agglomeration) being due to the lack of interplay between these two metals. It can be inferred that the introduction of polymers as bridges (adhesion) at the heterointerface is required. The EDS element mapping results of the Ag particles are shown in Figure 4b,c. Thus, improving the adhesion of Ag particles onto the surface of 316LSS is a crucial factor in retaining Ag particles after the subsequent sintering.

### 3.2. Introduction of Functional Polymers PEI-co-GA/Ag

Polyethyleneimine (PEI) is a hydrophilic polymer containing amine groups, including a large number of primary, secondary, and tertiary amines. Therefore, PEI is a highly branched cationic polymer, which makes it exhibit very good adhesion on the surface of the substrate [42,43,44]. According to Zoltán Géczi et al. (2018) [45], a Ag-PEI-PLA composite can be effectively used to coat the inner surface of acrylic dentures with antibacterial ability. However, the dispersion and adhesion of silver particles on metal substrates were not studied in any of the above-mentioned literature. In our study, the functional polymer, PEI-co-GA/Ag, was utilized innovatively for chemical silver plating on the 316LSS surface.

As shown in Figure 5a, the introduction of PEI-co-GA/Ag as a graft bridge can effectively enhance the adhesion between Ag and 316LSS surfaces. As can be seen in Figure 5b, the silver particles had a size of about 300–400 nm and were uniformly dispersed on the surface of 316LSS without agglomeration. Compared to 316LSS without the use of PEI-co-GA/Ag (see Figure 4a), a large number of Ag particles were retained on the surface of 316LSS after electroless silver plating. The EDS element mapping results of the Ag particles are shown in Figure 5c. Namely, the functional polymer, PEI-co-GA/Ag, can control the catalytic nuclei for electroless Ag deposition to achieve uniform dispersion and the presence of a certain attraction with the 3316LSS surface. The Ag particles of the functional polymer, PEI-co-GA/Ag, were further used as the activators for catalyzing the electroless silver. In general, the two different types of metal interfaces interact only through weak van der Waals forces, resulting in poor adhesion between them [46]. However, stronger hydrogen bonds can be formed between the functional polymer, PEI-GA-Ag, and the 316LSS surface [47,48,49]. Since PEI-co-GA/Ag contains a large number of amine groups, it can create hydrogen bonds with the oxides on the surface of stainless steel. The highly cross-linked structure of the PEI-GA-Ag composite is also capable of generating more van der Waals forces with the 316LSS surface [50]. Furthermore, the negative zeta potential of the metal surface is a critical parameter for attracting the functional polymer, PEI-co-GA/Ag (a cationic polymer), with a positive charge [51,52]. In that case, the charged cationic amine groups in the PEI structure are able to form a tight attraction with the surface of 316LSS while achieving an effective distribution of Ag NPs through the branched structure of the PEI. Based on this approach, the study successfully overcame the problem of uneven distribution and poor adhesion of the Ag NPs after sintering using Ag-containing stainless steel, which could not be achieved in the current literature. Figure 6a,b represents SEM images of raw 316LSS and 316LSS without the use of the functional polymer after the mirror silver reaction and sintering, respectively. Compared with the literature [30], as shown in Figure 6c, it can be seen that the Ag-containing 316LSS that had undergone electroless silver plating still retained a certain amount of Ag content (the white spots in the image), and the Ag NPs were well distributed after sintering. Additionally, inductively coupled plasma (ICP) was used to characterize and quantify the Ag content on the surface of the 316LSS.

The results further indicated that 316LSS, using the functional polymer PEI-co-GA/Ag, can effectively improve the retention of Ag NPs on metal substrates. Figure 7 displays the interaction of a schematic illustration between the functional polymer and the 316LSS surface.

### 3.3. Antibacterial Performance of 316LSS without/with PEI-co-GA/Ag

In the pilot study, two methods were introduced to examine the antibacterial ability and effectiveness of 316LSS with/without PEI-co-GA/Ag. First, the Kirby–Bauer Disk Diffusion Test [40,53] was used to examine whether the free silver ions were released. As shown in Figure 8a,b, two disks, 316LSS without PEI-co-GA/Ag and 316LSS with PEI-co-GA/Ag, which reacted for 4 h, were placed on a blood agar plate (Müller-Hinton Agar), spread with a streptococcus strain, and incubated in a 35 °C, 5% CO_2_ air incubator for 24 h. In Figure 8c, it can be seen that some bacterial strains appeared on the agar plate in contact with the surface of the unuse of the PEI-co-GA/Ag 316LSS. By contrast, the 316LSS with PEI-co-GA/Ag disk showed high antibacterial ability. It further showed no release of free silver ions to affect the surrounding environment (no inhibition zone around it). These results met our expectations of designing passive antimicrobial properties on the contact surface of medical devices.

Additionally, the Colony Forming Units (CFUs) were used to examine the antibacterial effectiveness of both disks. As shown in Figure 9a, the CFUs’ assay measures the number of viable microorganisms present in a sample [41,54]. First, we placed two disks, the 316LSS and the 316LSS PEI-co-GA/Ag, which reacted for 4 h, on the streptococcus blood agar plate for 24 h, and then we used a cotton swab to dip the surface of the two substrates for overnight bacterial culture. The result showed that the colonies of the control and the experimental groups were significantly different in Figure 9b,c, respectively. The average CFU/mL distribution of the antibacterial effect of 316LSS with/without PEI-co-GA/Ag is shown in Figure 9d and Table 1. Further experiments will evaluate the antibacterial effect of different Gram-positive and Gram-negative bacteria; in addition, biocompatibility and cytotoxicity studies will also be carefully examined.

## 4. Conclusions

In this study, the antimicrobial 316LSS sinter was successfully prepared by introducing the functional polymer PEI-co-GA/Ag composite in electroless silver plating. The amine group in the PEI structure can form a tight attraction with the surface of 316LSS while achieving an effective distribution of Ag NPs through the branched structure of the PEI. The use of PEI-co-GA/Ag composites to achieve good adhesion between Ag particles and the 316LSS surface can overcome the extremely low solubility of Ag in Fe and its precipitation at the grain boundaries during the sintering process. It can be seen from the SEM images that a large number of silver particles are retained and well dispersed in the 316LSS. At the same time, the probable mechanism for the influence of the functional composites on the enhancement of adhesion is also proposed. The formation of a large number of hydrogen bonds and van der Waals forces, as well as the negative zeta potential of the 316LSS surface, can effectively create a tight attraction between the copper layer and the 316LSS surface. More importantly, the antimicrobial 316LSS sinter exhibits highly effective antimicrobial effects by inhibiting bacterial proliferation on the contact surface of the material and does not release free silver ions to affect the surrounding environment. In summary, we have innovatively developed a technology to deposit Ag particles on the surface of 316LSS through electroless plating, which also applies to the subsequent sintering process. This method simplifies the preparation of antimicrobial stainless steel and has potential applications in orthopedic and dental implants.

## Figures and Tables

**Figure 1 materials-16-03719-f001:**
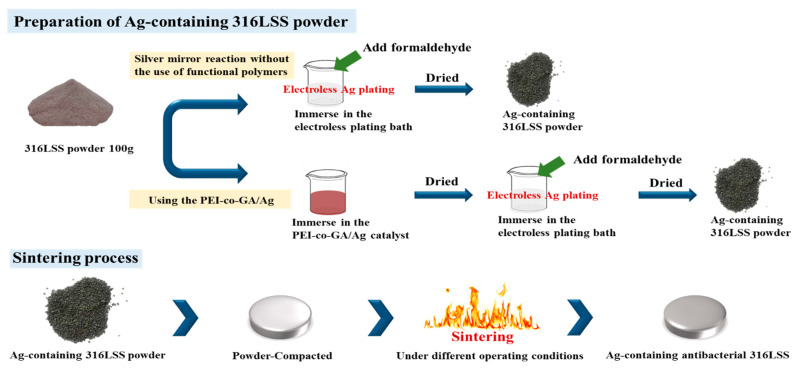
Schematic diagram of the experimental procedure for preparing Ag-containing antimicrobial 316LSS.

**Figure 2 materials-16-03719-f002:**
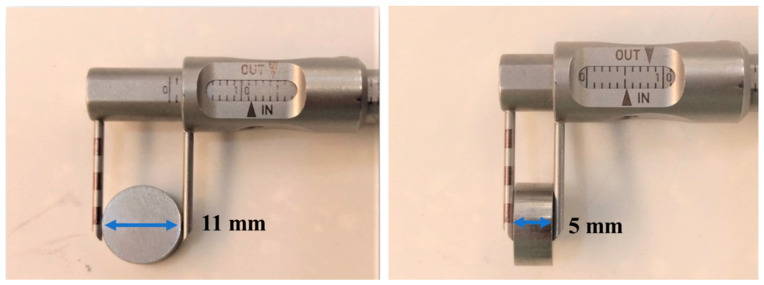
Powder-compacted dimensions with a diameter of 11 mm and a thickness of 5 mm.

**Figure 3 materials-16-03719-f003:**
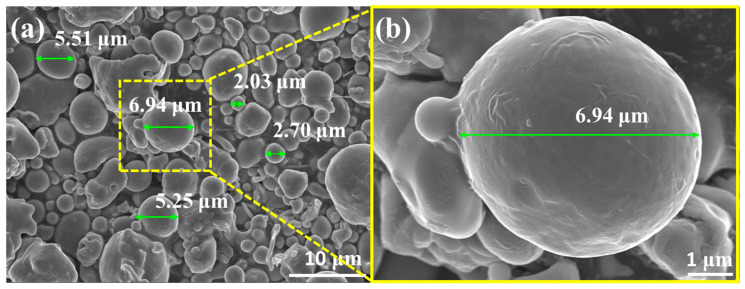
SEM images of raw 316LSS powders: (**a**) random particle size of about 2–7 μm, and (**b**) the surface morphology of the powder with high magnification.

**Figure 4 materials-16-03719-f004:**
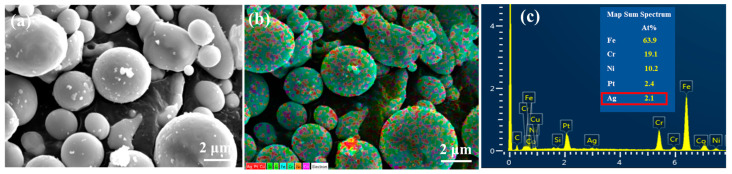
SEM images of raw 316LSS powders after the silver mirror reaction without the use of functional polymers: (**a**) the surface morphology of the powders, and (**b**,**c**) the EDS element mapping results.

**Figure 5 materials-16-03719-f005:**
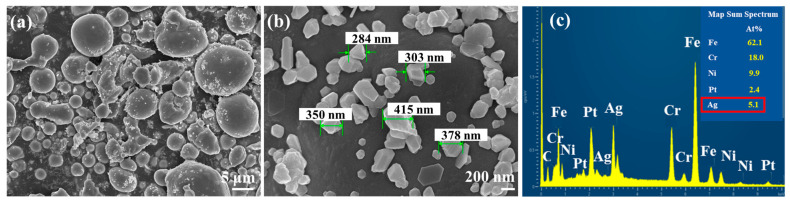
SEM images of 316LSS using the PEI-co-GA/Ag composite after electroless silver plating: (**a**) the surface morphology of the powders, (**b**) a particle size of about 300–400 nm, and (**c**) the EDS element mapping results.

**Figure 6 materials-16-03719-f006:**
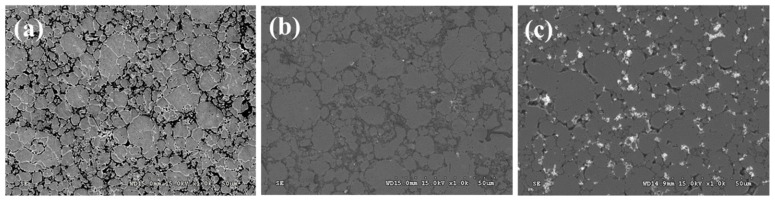
SEM images of 316LSS after electroless silver plating and sintering: (**a**) the surface morphology of raw 316LSS; (**b**) the 316LSS without the use of the functional polymer showed little retention of Ag NPs; (**c**) the 316LSS with the functional polymer showed a certain amount of Ag content (white spots) and an effective distribution of Ag NPs.

**Figure 7 materials-16-03719-f007:**
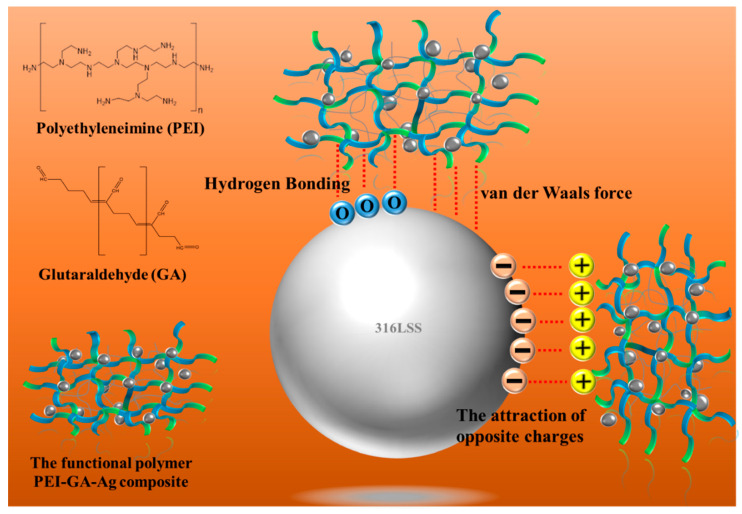
Schematic illustration of the interaction of the functional polymer PEI-co-GA/Ag composite utilized in this study with 316LSS.

**Figure 8 materials-16-03719-f008:**
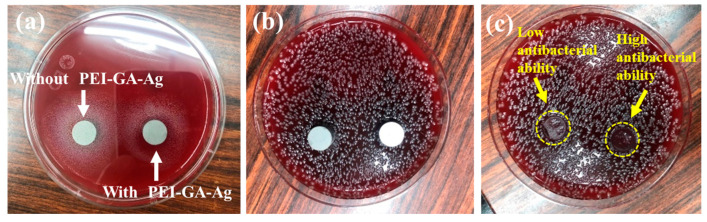
Kirby–Bauer disk diffusion test: (**a**) two disks, 316LSS without PEI-co-GA/Ag (Left) and 316LSS with PEI-co-GA/Ag (Right), reacted for 4 h, were placed on a blood agar plate which was covered with a streptococcus strain for 24 h. (**b**) The agar plate was placed in a 35 °C, 5% CO_2_ air incubator after 24 h. (**c**) We removed the test disks with forceps after 24 h; the contact surface of the left disk shows a small amount of bacteria strain, and the contact surface of the right disk did not grow a bacteria strain.

**Figure 9 materials-16-03719-f009:**
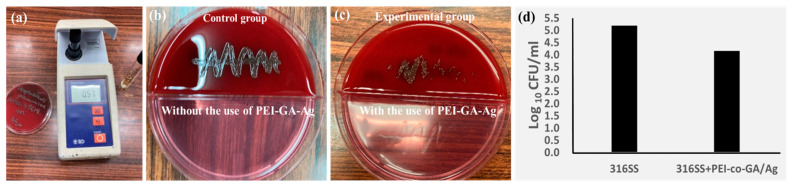
Colony Forming Units’ (CFUs’) assay (**a**) Bacteria strain: Streptococcus pneumoniae, ATCC49619 OD600:0.57, was used for the antibacterial test. (**b**) 316LSS without PEI-GA-Ag (**c**) 316LSS with PEI-GA-Ag, reaction time 4 h. (**d**) The data of CFU show a significant difference between 316LSS and 316LSS with PEI-GA-Ag.

**Table 1 materials-16-03719-t001:** The average CFU/mL distribution of the antibacterial effect of 316LSS with/without the use of PEI-co-GA/Ag.

	CFU COUNT 1	CFU COUNT 2	CFU COUNT 3	Mean	LOG	UP	Down
316SS	156,000	165,000	134,000	151,667	5.18	0.037	0.05
316SS + PEI-co-GA/Ag	10,500	12,300	19,950	14,250	4.15	0.146	0.13

## Data Availability

The data presented in this study are available on request from the corresponding authors.

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
