# Peer review of "A Novel Method for the Fabrication of Antibacterial Stainless Steel with Uniform Silver Dispersions by Silver Nanoparticle/Polyethyleneimine Composites"

_materials, 2023, doi:10.3390/ma16103719_

Round 1

Reviewer 1 Report

this manuscript is not of the highest scientific quality considering the journal's targe. It is rather a technical report, full of pictures 

Only SEM and a few unrepresentative EDS 

The antibacterial test is the only interesting aspect, but it is not described and there is no statistical analysis

Reviewer 2 Report

The research article entitled "A novel method for the fabrication of antibacterial stainless 2 steel with uniform silver dispersions by silver nanoparticle- 3 /Polyethyleneimine composites" by Yu-Kun Chih deals with the fabrication of stainless steel with polymer. The objective of the article was very clear and explained the concept with different characterization techniques. 

From my point of view, I suggest that the research article can be published in the journal without any further modification.

Reviewer 3 Report

In this manuscript entitled “A novel method for the fabrication of antibacterial stainless 2 steel with uniform silver dispersions by silver nanoparticle/Polyethyleneimine composites” (Manuscript Number: materials-2306333) I think it’s better to discuss about below questions. Therefore, I suggest a major revision for the manuscript before publication.

Comments:

  1. Abstract: should be start with brief introduction.
  2. Abstract: should be presented with more results.
  3. Experimental: should be extended. Also, the total materials and companies, characterization of all instruments, methodology, and preparation steps should be presented with more details.
  4. The text can be improved by providing a more critical discussion of related literature. For example: Microporous and Mesoporous Materials 218, 62-68 (2015) / RSC Advances 5 (19), 14407-14415 (2015).
  5. Conclusions: should be extended.
  6. The authors should be explaining about the importance and novelty of the work with more details.
  7. In the text there are some grammatical and syntactic errors that should be corrected. For example "by" or "using" not "by using" and etc.
  8. The English language should be improved.

Reviewer 4 Report

This manuscript describes the fabrication of antibacterial stainless steel by introducing silver nanoparticles into functional polymers and clarifies the material properties and antibacterial effect. However, this manuscript would be not acceptable for publication without revising the points itemized below.

1. It is necessary to add the preparation procedures of the 316LSS powders after silver mirror reaction without the use of functional polymers in Fig. 4, and the 316LSS after electroless silver plating and sintering without the use of functional polymer in Fig. 6(b) in 2.2 part of Material and methods.

2. The evaluation methods of antibacterial activity should be added to Material and methods.

3. To clarify the effect on introducing the functional polymer PEI-co-GA, antibacterial activity should be investigated for three materials (Fig.6(a), (b), (c)): 316LSS only, 316LSS/Ag with and without the use of functional polymer.

4. In Table 1, the unit of colony counts should be revised to CFU/m2, because the surface of the 316LSS is effective against the bacteria.
